Feeding ecology and trophic interactions of the narrow-barred Spanish Mackerel (Scomberomorus commerson) in the Central Taiwan Strait

Cheng Li Chi 1
He Jia Shin 1
Lai Chi Chang 1
Lee Yen Hung 2
Weng Jinn Shing 1 j-s.ueng@mail.tfrin.gov.tw
Huang Hsing Han 1
Wu Yi Shu 1
1 Coastal and Offshore Fishery Research Center, Fisheries Research Institute, Ministry of Agriculture , Kaohsiung City , Taiwan
2 Tungkang Aquaculture Research Center, Fisheries Research Institute, Ministry of Agriculture , Pingtung City , Taiwan
Yapıcı Sercan
Electronic publication date: 2025 Nov 14
Publication date: 2025
Volume: 13
Electronic Location ID: e20350
Received 2025 May 7; Accepted 2025 Oct 15
Copyright: © 2025 Cheng et al.
Copyright year: 2025
Copyright holder: Cheng et al.
License: This is an open access article distributed under the terms of the Creative Commons Attribution License, which permits unrestricted use, distribution, reproduction and adaptation in any medium and for any purpose provided that it is properly attributed. For attribution, the original author(s), title, publication source (PeerJ) and either DOI or URL of the article must be cited.
License URL: https://creativecommons.org/licenses/by/4.0/

Keywords: Scomberomorus commerson, Diet composition, Stomach content analysis, Stable isotope analysis, Isotopic mixing model, Trophic position

Funding: Ministry of Agriculture, Taiwan (R.O.C) 113AS-6.5.1-AI-01, 114AS-6.4.1-AI-04 This study was financially supported by the Ministry of Agriculture, Taiwan (R.O.C) on projects 113AS-6.5.1-AI-01 and 114AS-6.4.1-AI-04. The funders had no role in study design, data collection and analysis, decision to publish, or preparation of the manuscript.

==============================
Understanding the foraging ecology of marine predators is essential for ecosystem-based fisheries management. This study examined the diet of the narrow-barred Spanish mackerel (Scomberomorus commerson) in the Central Taiwan Strait using stomach content and stable isotope analyses integrated with an isotopic mixing model. A total of 1,733 specimens were collected between January 2017 and March 2022. Stomach content analysis revealed that 79.8% had empty stomachs, while the remainder mainly contained semi-digested fish remains. Among identifiable prey, composition varied by size and season. Sardinella lemuru, Decapterus spp., and Trichiurus spp. dominated the diet of smaller individuals, while larger fish primarily consumed Decapterus spp. and Mene maculata. Sardinella lemuru and Trichiurus spp. were more abundant in summer and autumn–winter, respectively. In the meanwhile, Decapterus spp. remained important year-round. Isotopic analysis further revealed that Sardinella lemuru, Etrumeus micropus, Decapterus macarellus and Penaeidae (Metapenaeopsis barbata) were dominant prey in smaller size classes, while Encrasicholina punctifer, D. macarellus, Evynnis cardinalis, Trichiurus spp. and Uroteuthis spp. were prevalent in intermediate and larger individuals. The estimated trophic position (3.8–4.3; mean = 4.0) confirms that S. commerson functions as a top predator with opportunistic feeding behavior. These findings improve understanding of regional trophic dynamics and support sustainable fisheries management in the Central Taiwan Strait.

Introduction

Pelagic predators play pivotal roles in marine ecosystems by regulating prey populations and mediating energy transfer across trophic levels (Bornatowski et al., 2018; Grainger et al., 2020; Stevens et al., 2000; Tran et al., 2023). Their foraging strategies and trophic dynamics are also of major interest to fisheries management, particularly for species that support valuable commercial harvests (Navia et al., 2012; Longo et al., 2015).

The narrow-barred Spanish mackerel (Scomberomorus commerson) is a pelagic species that is widely distributed across the Indo-Pacific, from the Red Sea and the waters off South Africa to the seas off various regions in Asia and Oceania, including Taiwan, Japan, and Australia (Randall & Hoover, 1995). This species typically inhabits waters along the edges of the continental shelf and shallow areas at depths of <100 m (Niamaimandi et al., 2015). It is known for its extensive migration, which often exceeds 1,000 nautical miles (McPherson, 1989; Siddeek, 1995; Rohit & Abdussamad, 2013). Scomberomorus spp. was found occupy a relatively high trophic level (~4.35), placing them the top predators in the Central Taiwan Strait (Wu et al., 2024). Along the southwest coast of India, studies have shown that the diet of S. commerson mainly consists of fish (99.9%), with crustaceans as a secondary component (Rajesh et al., 2017). Stomach content analyses (SCAs) have further revealed that the diet composition of S. commerson varies according to prey availability and the size relationship between predator and prey (Harden, Alexander & Kennedy, 1967; Kakuda & Matsumoto, 1978; Rubin & Davis, 1985; Rohit & Abdussamad, 2013).

While the trophic dynamics and diet composition of Scomberomorus spp. have been investigated in parts of the Pacific (Dambacher et al., 2010), such information remains understudied in Taiwanese waters. Previous research has primarily focused on its age, growth, reproductive biology, and migratory behavior (Chen, 1973; Lo, 2018; Weng et al., 2020, 2021).

According to Lo (2018), the migrations of S. commerson closely align with the 23–26 °C isotherm, reflecting seasonal movements in response to seasonal current patterns. In spring, individuals migrate northward from the southern Taiwan Strait, including the Taiwan Bank, western Taiwan, and the southeastern coast of China, as the influence of the cold, nutrient-rich China Coastal Current diminishes. During summer, the species continues northward into the central and northeastern Taiwan Strait or the East China Sea, where the warm, oligotrophic Taiwan Strait Current and South China Sea Warm Current prevail. In late autumn and winter, S. commerson migrated southward again as the China Coastal Current intensified and seawater temperatures decreased, with distributions centered around the southwestern coast of Taiwan, the Taiwan Bank, and the northern South China Sea (Fig. 1A).

Figure 1 Seasonal migration patterns of S. commerson in the northwestern Pacific (A) and fishing areas of specimens (n = 1,733) collected in this study (B).

(A) Seasonal migration routes (modified from Lo (2018)). Colors indicate migration periods: spring (green), summer (red), autumn (orange), and winter (blue). (B) Sampling locations in the central Taiwan Strait, with fishing gear types used for specimen collection: longline (white), trolling line (light grey), driftnet (cross-hatched), light fishery (star), and trawling net (dark grey square).

Reproductive biology further highlights the seasonal ecology of the species in Taiwanese water. Weng et al. (2020) reported that spawning occurs from March to August, peaking between March and May. More recently, Weng et al. (2021) demonstrated that individuals currently caught in the central Taiwan Strait are primarily aged 1+ to 2+ years. Estimates of fishing mortality (0.27 y−1) and exploitation rate (0.30) suggest that overfishing is not presently occurring in this stock. Nevertheless, S. commerson commands high market value and supports intensive fisheries throughout the region. In Taiwan, there are no regulations governing the number or size of individuals landed by traditional fisheries. Although the stock is not considered overexploited in the Taiwan Strait (Weng et al., 2021), official statistics indicate a marked decline in catch, from a peak of 6,600 t in 2002 to 1,238 t in 2023 (Fisheries Agency, 2025). This apparent discrepancy highlights the need for comprehensive biological and ecological studies to inform stock assessments and resource management. In particular, the feeding ecology and trophic interactions of S. commerson in Taiwanese waters remain insufficiently understood, warranting further investigation.

Dietary studies of marine vertebrates often rely on SCA, which provides direct information on prey categories and their proportions (Hyslop, 1980). However, SCA has limitations, including potential underestimation of certain prey items due to differential digestion rates (MacDonald, Waiwood & Green, 1982). By contrast, stable isotope analysis (SIA) can help determine isotope turnover in predator tissues, offering insights into long-term dietary patterns.

Metabolic activity influences isotope turnover rates across tissues, which in turn affect stable isotope values (Fry & Arnold, 1982). The muscle tissue below the dorsal fin shows lower variance in δ15N and δ13C values. It is therefore widely used to reconstruct feeding patterns over periods ranging from several months to a year (Madigan et al., 2012). This tissue has also proven useful for evaluating food web structures (Pinnegar & Polunin, 1999; Deudero et al., 2004).

SIA can further clarify the community structure and trophic dynamics by providing quantitative estimates of trophic positions (TPs) and individual specialization (Peterson & Fry, 1987; Vander Zanden & Rasmussen, 2001). Changes in δ15N and δ13C values during digestion and assimilation reflect isotope fractionation, which can reveal shifts in TPs (Minagawa & Wada, 1984). For instance, a 2.96–3.4‰ increase in δ15N corresponds to a rise in TP (Vanderklift & Ponsard, 2003).

The combination of SCA and SIA, along with an isotopic mixing model offers a powerful approach for reconstructing the ecological niches and foraging patterns of pelagic predators (Harvey et al., 2002). While this integrative framework has been used to examine tuna species in Taiwanese waters (Weng et al., 2015; Chang et al., 2022), it has rarely been adopted for S. commerson in the study area. Therefore, this study aims to investigate the trophic interactions and ontogenetic dietary shifts of S. commerson in the Taiwan Strait using these complementary methods. By integrating temporal and size-related dietary data, we clarify its role in the pelagic food web of the northwestern Pacific. Our findings may contribute to the development of ecosystem-based fisheries management strategies.

Materials and Methods

Sampling

A total of 1,733 specimens of S. commerson (fork length [FL]: 21.5–159 cm) were collected monthly from the Penghu fish market between January 2017 and March 2022. To account for gear-related limitations in sample collection, most specimens were caught using driftnets, longlines, and trolling lines operated by commercial vessels in the central Taiwan Strait (Fig. 1B). However, individuals with a FL < 50 cm were primarily collected using light fisheries and trawling nets from vessels also operating in the study area (Fig. 1B). This approach ensured better representation of smaller size classes for subsequent analyses. Immediately after capture, the specimens were stored on ice and transported to the laboratory for analysis.

Stomach content analysis (SCA)

For each specimen, FL (to the nearest 0.1 cm) and body weight (to the nearest 0.01 kg) were recorded. Stomachs were dissected, and prey items were identified to the lowest possible taxonomic level. Prey species, numbers, and weights were recorded. The diet composition of S. commerson is expressed using the following indices: (a) Weight percentage (%W): proportion of each prey type by weight relative to the total stomach content,

(b) Numerical percentage (%N): proportion of each prey type by count relative to the total stomach content,

(c) Frequency of occurrence (%F): percentage of non-empty stomachs containing each prey type, and

(d) Relative importance index (%IRI): [(%Ni + %Wi) × %Fi]/ ∑i=1a⁡[(%Ni+%Wi)×%Fi] × 100%, where “i” represents specific prey and “a” represents total prey (Pinkas, Oliphant & Iverson, 1970; Hyslop, 1980).

Stable isotope analysis (SIA)

A subset of S. commerson specimens (n = 152) was randomly selected for SIA. White muscle tissue samples were collected from below the second dorsal fin. Additional tissue samples from copepods and prey species known to constitute the diet of S. commerson, based on SCA results and previous studies (n = 94), including fish, shrimp, and cephalopods, were obtained either from the stomach contents of the specimens or from fish markets. All samples were washed with deionized water, dried at 60 °C for 24 h, and ground into homogenized powder using an agate mortar and pestle. For isotope analysis, 0.7–0.75 mg of each sample was encapsulated in an 8 mm × 5 mm tin cup.

Following the protocols of Post et al. (2007), Logan et al. (2008), and Skinner, Martin & Moore (2016), lipid extraction was deemed unnecessary because the mean C:Nbulk ratio of S. commerson white muscle tissue was 3.3 ± 0.52, which is below the commonly accepted threshold of 3.5, indicating minimal lipid influence on isotopic values.

The δ15N and δ13C values of the samples were measured using a Flash 2000 automatic elemental analyzer coupled to a Finnigan MAT 253 isotope ratio mass spectrometer (Thermo Finnigan, Egelsbach, Germany) at the Institute of Oceanography, National Taiwan University. To ensure accuracy, standards such as urea, protein, and USGS40 (L-glutamic acid) were analyzed after every seven samples. Stable isotope values are expressed in δ-notation as follows:

δX=[(Rsample/Rstandard)−1]×103(%)

where X represents 15N or 13C and R represents the 15N:14N or 13C:12C ratio in the sample relative to the standard (Peterson & Fry, 1987). PeeDee belemnite and atmospheric nitrogen were used as reference standards. Analytical precision was set at ±0.15‰ for both carbon and nitrogen isotopes.

Trophic enrichment factors (TEF) and trophic position (TP) determination

The trophic enrichment factor (TEF) represents the difference in stable isotope ratios between a consumer and its dietary sources (Dionne, Dufresne & Nozais, 2016), expressed in Δ-notation as follows: ΔX (‰) = δXconsumer−δXprey species, where δX refers to either δ15N or δ13C values of the consumer (S. commerson) and its prey in this study. The carbon TEF is denoted as Δδ13C, and the mean Δδ13C across all of common prey items is referred to as Δδ13Cc. For nitrogen, a general Δ15N value of 3.4‰ was adopted for marine fishes (Vander Zanden & Rasmussen, 1999; Post, 2002), hereafter referred to as Δ15Ngeneral. Additionally, the nitrogen TEF was calculated based on the empirically measured isotopic values of common prey species in this study, denoted as Δ15Nanalyzed. The mean value of Δ15Nanalyzed across all prey items was defined as Δ15Nc and was subsequently used for estimating the trophic position (TP) of S. commerson.

TP was determined using the following equation (Vander Zanden & Rasmussen, 1999; Post, 2002; Aberle & Malzahn, 2007; Dambacher et al., 2010):

TP=δ15Nconsumers−δ15NbaselineTEF+λ

where δ15Nconsumers represents the δ15N value of S. commerson and λ represents the TP of marine primary consumers, which was set at 2 (e.g., herbivorous fish and zooplankton). The δ15Nbaseline value was derived from zooplankton samples collected from Taiwanese waters (copepods, 6.3 ± 0.7‰, n = 5). Two TEF values were used for TP estimation: Δ15Ngeneral (3.4‰) and Δ15Nc, with the resulting trophic positions referred to as TPgeneral and TPc, respectively, for further comparison.

Statistical analysis

To assess variations in SCA and SIA results, S. commerson specimens were categorized by size and season. Following the approach of Weng et al. (2015), we categorized the specimens into six size classes with different FLs: <50, 50–70, 70–90, 90–110, 110–130, and >130 cm. Seasons were defined as spring (March–May), summer (June–August), autumn (September–November), and winter (December–February of the following year).

The Kruskal–Wallis one-way ANOVA (non-parametric) was used to test for differences in δ15N and δ13C values of specimens among size classes and seasons, respectively. All tests were two-tailed and conducted in MATLAB R2020b (The MathWorks Inc, 2020). For each test, the Chi-square statistic and corresponding p-value were reported, with a significance threshold of α = 0.05. Post hoc multiple comparisons were subsequently performed in MATLAB R2020b (The MathWorks Inc, 2020) to determine which groups differed significantly.

To estimate the relative contributions of different prey species to each size class, isotope values and TEFs for S. commerson and its common prey were analyzed using the Bayesian isotopic mixing model MixSIAR version 3.1.12 (University of California, Santa Cruz, California, USA) (Parnell, Inger & Bearhop, 2010; Stock & Semmens, 2016).

The isotope data of S. commerson were grouped by size class using the “pack” parameter as a fixed effect. TEFs (Δ15Nc = 3.72 ± 1.5 ‰, Δ13Cc = 0.2 ± 0.98‰) were calculated as the mean difference between the stable isotope values of S. commerson muscle tissue and those of its potential prey, based on empirically derived values obtained in this study.

A residual × process error structure was applied to account for both observation error and ecological variability. The Markov chain Monte Carlo (MCMC) parameters were set to three chains of 300,000 iterations each, with a burn-in of 200,000 iterations and a thinning interval of 100, yielding 3,000 posterior samples per chain (9,000 samples in total). Model convergence was evaluated using Gelman–Rubin and Geweke diagnostics, both of which indicated satisfactory convergence (all Gelman–Rubin statistics <1.05).

To address ontogenetic dietary variation, a two-step modeling approach was used. In the first run, all common prey groups were included; those contributing <5% with 95% credibility intervals overlapping 0 were excluded. The reduced set of prey sources was then used in a second run to re-estimate dietary contributions for each size class using the same model settings. The outputs from this model informed our understanding of trophic interactions and ontogenetic dietary shifts in S. commerson.

Results

SCA results

Of the 1,733 specimens, 1,383 (79.80%) had empty stomachs. The distribution of size classes and seasonal feeding incidence across five fishing methods is presented in Fig. 2. Most specimens were caught using drift gillnets (56.61%) and longlines (27.35%), with individuals in Classes III and IV dominating across all seasons. For driftnet and longline fisheries, seasonal feeding incidence ranged from 10.53% to 26.63%. In the case of trolling lines, the highest feeding incidence (50%) was recorded in autumn, with values ranging from 13.79% to 33.33% in other seasons. Among small individuals caught by light fisheries and trawling nets, feeding incidence ranged from 53.85% to 62.5%, except for one individual collected by trawling in summer, which had an empty stomach. Overall, feeding incidence varied by season, size class, and fishing method, with generally lower values observed in winter.

Figure 2 Seasonal, size composition and feeding incidence (%) of S. commerson by five fishing methods (n = 1,733; driftnet = 981, longline = 474, trolling net = 248, light fishery = 16, trawling net = 14).

Stacked bars represent the size class composition across seasons for each fishing method, with sample sizes (n) labeled above each bar. Feeding incidence (%) is shown as a line plot with circular markers and is scaled to the right Y-axis.

Dietary analysis was conducted for the remaining 350 specimens (20.20%, Fig. 3). The diet of S. commerson was predominantly composed of semi-digested fish remains, with %W, %N, and %IRI values of 57.14%, 39.95%, and 88%, respectively. To assess variations in diet composition across the size classes and seasons, prey species were categorized into 15 groups. Feeding incidence (%) and stomach content composition, based on prey item counts, are summarized in Tables 1 and 2.

Figure 3 Diet composition in stomach contents of S. commerson (n = 350) stratified by weight percentage (%W), numerical percentage (%N), frequency of occurrence (%F), and relative importance index (%IRI).

Table 1 Feeding incidence and stomach content composition of S. commerson specimens, categorized by size class, based on the number of prey species identified.

	Size classes	
Class I
(<50 cm)	Class II
(50–70 cm)	Class III
(70–90 cm)	Class IV
(90–110 cm)	Class V
(110–130 cm)	Class VI
(≧130 cm)	Total	
Size range (cm)	21.5–49.5	52.1–69.8	70–89.5	90–109.5	110–129.5	130–159		
Mean length (cm)	37.34 ± 9.67	64.38 ± 5.67	81.92 ± 6.33	97.83 ± 5.24	114.75 ± 4.88	137.66 ± 6.78		
No. of specimens examined	30	26	318	1,128	212	19	1,733	
No. of specimens feeding	17	8	52	225	43	5	350	
Feeding incidence (%)	56.67	30.77	16.35	19.95	20.28	26.32	20.20	
Contents (n)								
Fish								
Encrasicholina spp.			31		5		36	
Sardinella lemuru	17			1	11		29	
Saurida spp.			1	2			3	
Decapterus maruadsi			2	4	1		7	
Decapterus spp.	2		12	37	10	1	62	
Mene maculata				1		2	3	
Evynnis cardinalis			1	1			2	
Ammodytidae			1				1	
Scomber spp.			1	2			3	
Katsuwonus spp.				1	1		2	
Trichiurus spp.		2	1	5	2		10	
Cephalopod								
Uroteuthis spp.		1	3	11	2	1	18	
Sepioteuthis lessoniana				1			1	
Digested remains								
Semi-digested shrimp remains	1			2			3	
Semi-digested fish remains	7	6	37	159	28	3	240	

Table 2 Feeding incidence and stomach content composition of S. commerson specimens, categorized by seasons, based on the number of prey species identified.

	Spring	Summer	Autumn	Winter	
Size range (cm)	65.5–159	21.5–142.5	42.5–139	71–143	
Mean length (cm)	99.99 ± 10.96	77.43 ± 26.05	87.96 ± 19.67	96.77 ± 8.73	
No. of specimens examined	718	139	135	741	
No. of specimens feeding	151	35	39	125	
Feeding incidence (%)	21.03	25.18	28.89	16.87	
Contents (n)					
Fishes					
Encrasicholina spp.	26			10	
Sardinella lemuru	11	14	4		
Saurida spp.	2			1	
Decapterus maruadsi	5	1		1	
Decapterus spp.	34		78	13	
Mene maculata	2	1			
Evynnis cardinalis		1	1		
Ammodytidae		1			
Scomber spp.		1	1	1	
Katsuwonus spp.	1			1	
Trichiurus spp.			3	7	
Cephalopod					
Uroteuthis spp.	4	1	3	10	
Sepioteuthis lessoniana			1		
Digested remains					
Semi-digested shrimp remains			2	1	
Semi-digested fish remains	100	19	23	98	

Variations in diet composition across size classes

The primary contributors to %W (Fig. 4) in Class I were species from Sardinella lemuru (52.90%) and semi-digested fish remains (36.85%). Trichiurus spp. (88.84%) and semi-digested fish remains (10.69%) were predominant in Class II. In Class III, semi-digested fish remains (37.80%) and Decapterus spp. (37.01%) were the major contributors. Similarly, Class IV mainly contained semi-digested fish remains and Decapterus spp. (45.35% and 23.76%, respectively). Class V contained Decapterus spp. (37.17%), semi-digested fish remains (28.78%), and Trichiurus spp. (26.77%). In Class VI, Mene maculata (48.82%) and Decapterus spp. (34.82%) were dominant.

Figure 4 Prey composition in stomach contents by size class (I–VI) in S. commerson (n = 350; Class I = 17, II = 8, III = 52, IV = 225, V = 43, VI = 5).

Data are presented as weight percentage (%W), numerical percentage (%N), frequency of occurrence (%F), and relative importance index (%IRI).

Similar trends were observed for number-based contributions (%N, Fig. 4). Class I mainly contained species from Sardinella lemuru (41.18%), semi-digested fish remains (41.18%), and Decapterus spp. (11.76%). The primary prey in Class II were semi-digested fish remains (66.67%) and Trichiurus spp. (22.22%). Class III mostly consisted of semi-digested fish remains (62.71%) and Decapterus spp. (20.34%). Similarly, Class IV mainly contained semi-digested fish remains (66.81%), followed by Decapterus spp. (15.55%) and Encrasicholina spp. (5.04%). The primary prey in Class V were semi-digested fish remains (60.87%) and Decapterus spp. (21.74%). A more diverse dietary pattern was noted for Class VI, which contained semi-digested fish remains (42.86%), Mene maculata (28.57%), Decapterus spp. (14.29%), and Uroteuthis spp. (14.29%).

A similar pattern was observed for %F (Fig. 4). Consistently, semi-digested fish remains were the most frequently observed items across Class II to VI. The %F of semi-digested fish remains ranged from 25.93% in Class I to 66.67% in Class II, indicating its high contribution. Other notable prey items in the size classes included Sardinella lemuru (62.96% and 18.33% in Classes I and V, respectively), Trichiurus spp. (22.22% in Class II), Decapterus spp. (14.34–20.34% in Classes III–V), and Mene maculata (28.57% in Class VI).

These findings were further confirmed by %IRI (Fig. 4). Semi-digested fish remains were the most important dietary component across all size classes, ranging from 34.08% in Class I to 90.67% in Class IV. Sardinella lemuru (62.91%) were the primary prey items in Class I. Trichiurus spp. (31.83%) and Mene maculata (39.43%) were the key contributors in Classes II and VI, respectively. In Classes III–V, Decapterus spp. were consistently predominant, with the %IRI values of 15.37% in Class III and 19.57% in Class V. Thus, Decapterus spp. were important dietary components across these size classes.

Seasonal variations in diet composition

Seasonal variations were noted in diet composition (Fig. 5). In spring, the primary contributors to %W were Decapterus spp. (38.17%) and semi-digested fish remains (30.73%). In summer, Decapterus spp. (44.91%) were the dominant dietary components, followed by semi-digested fish remains (15.16%) and Mene maculata (11.53%). In autumn and winter, the diets were characterized by semi-digested fish remains (35.58–55.31%), Decapterus spp. (12.89–28.05%), and Trichiurus spp. (19.32–20%).

Figure 5 Seasonal variations in prey composition of S. commerson (n = 350; spring = 151, summer = 35, autumn = 39, winter = 125).

Data are presented as weight percentage (%W), numerical percentage (%N), frequency of occurrence (%F), and relative importance index (%IRI) values.

The results for number-based contributions (%N, Fig. 5) revealed that in spring, semi-digested fish remains (63.29%) and Decapterus spp. (21.52%) were the main dietary components. Similar trends were observed in summer and autumn. Semi-digested fish remains (51.11–51.35%), Decapterus spp. (17.78–18.92%), and Sardinella lemuru (6.67–13.51%) were the major dietary components. In winter, semi-digested fish remains dominated the diet (72.06%), followed by Decapterus spp. (9.56%) and Uroteuthis spp. (7.35%).

Similar trends were noted for %F across seasons (Fig. 5). In spring, semi-digested fish remains exhibited the highest %F (54.05%), followed by Decapterus spp. (18.38%) and Encrasicholina spp. (14.05%). In summer and autumn, consistently, the diets mainly consisted of semi-digested fish remains (41.30–50%), Decapterus spp. (15.22–17.39%), and Sardinella lemuru (8.7–30.43%). In winter, semi-digested fish remains (68.53%) remained the dominant component, followed by Decapterus spp. (9.09%), Encrasicholina spp. and Uroteuthis spp. (6.99%).

Semi-digested fish remains were the most significant dietary component across all seasons, with %IRI values ranging from 62.06% to 95.52% (Fig. 5). The %IRI values for Decapterus spp. ranged from 2.25% to 24.35%. These findings suggest that although the diet composition of S. commerson exhibits some seasonal variation, semi-digested fish remains are the predominant dietary component throughout the year.

SIA results

SIA results for S. commerson

For S. commerson (n = 152; FL: 90.04 ± 21.28 cm), the mean δ15N value in muscle tissue was 13.6 ± 0.6‰ (range: 12.0‰ to 15.3‰; Fig. 6). The mean δ13C value was −17.1 ± 1.2‰ (range: −20.2‰ to −15.5‰; Fig. 6). Both δ15N and δ13C values varied significantly across size classes (δ15N: χ2 = 29.14, p < 0.001; δ13C: χ2 = 18.78, p = 0.002).

Figure 6 Mean δ15N and δ13C values (the lines represent corresponding standard deviations) for S. commerson and its common prey species in the Central Taiwan Strait.

Sample sizes (n) is provided in the legend.

The mean δ15N values across size classes (Table 3) ranged from 12.6 ± 0.5‰ to 14.0 ± 1.0‰. The difference between Classes I and II was 0.72‰, followed by smaller incremental increases of 0.2–0.7‰ across the larger classes. Substantial within-class variations (2.5–2.9‰) were observed in Classes IV, V, and VI. Overall, δ15N values in Class I were significantly lower than those in Classes III, IV, V, and VI (p < 0.005, Table 3). A weak but significant positive correlation was found between FL and δ15N values (δ15N = 12.881 + 0.0036 FL; r = 0.342; p < 0.001; Table 3; Fig. 7A).

Table 3 δ15N and δ13C values (‰) in the muscle tissue and estimated trophic position (TPgeneral, TPc) of S. commerson specimens across size classes.

Size class	n	Mean FL ± SD	Mean δ15N ± SD	Mean δ13C ± SD	Mean TPgeneral ± SD	Mean TPc ± SD*	
I	9	36.82 ± 4.24	12.6 ± 0.5a	−16.6 ± 0.6a, b	3.8 ± 0.1	3.7 ± 0.1	
II	11	62.64 ± 4.49	13.3 ± 0.5a,b	−16.0 ± 0.3a	4.1 ± 0.2	3.9 ± 0.1	
III	48	82.32 ± 5.29	13.8 ± 0.4b	−17.2 ± 1.1b	4.2 ± 0.1	4.0 ± 0.1	
IV	67	98.58 ± 5.84	13.5 ± 0.5b	−17.2 ± 1.3b	4.1 ± 0.1	3.9 ± 0.1	
V	12	116.13 ± 6.00	13.8 ± 0.7b	−17.6 ± 1.0b	4.2 ± 0.2	4.0 ± 0.2	
VI	5	143.10 ± 9.53	14.0 ± 1.0b	−17.1 ± 1.6b	4.3 ± 0.3	4.1 ± 0.3	
Total	152	90.04 ± 21.28	13.6 ± 0.6	−17.1 ± 1.2	4.1 ± 0.2	4.0 ± 0.2	
Note:

δ15N and δ13C values followed by different superscript letters (a, b) indicate statistically significant differences among size classes (Kruskal–Wallis test with post hoc multiple comparison, p < 0.05).

Figure 7 Relationships between fork length and δ15N (A) and δ13C (B) values in S. commerson across all size classes (n = 152, Class I = 9, II = 11, III = 48, IV = 67, V = 12, VI = 5).

The solid line denotes linear regression, whereas the dotted line represents 95% confidence intervals.

Higher δ13C values were observed in smaller size classes (−16.6 ± 0.6‰ and −16.0 ± 0.3‰ in Classes I and II, respectively, Table 3). δ13C values slightly decreased to −17.6‰ to −17.1‰ in Classes III to VI. Greater individual variability (4.5–4.6‰) was observed in Classes III to IV, whereas the lowest within-class variation (1.0‰) in δ13C values was observed in Class II. δ13C values in Class II were significantly different from those in Classes III, IV, and V (p < 0.005, Table 3). A weak but significant negative correlation was observed between FL and δ13C (δ13C = −15.925 − 0.0134 FL; r = −0.176; p = 0.03; Table 3; Fig. 7B).

Similar to the aforementioned variations across size classes, significant seasonal variations were noted in δ15N and δ13C values (δ15N: χ2 = 10.78, p = 0.013; δ13C: χ2 = 9.6, p = 0.022, Table 4). The mean δ15N values declined slightly from 13.8 ± 0.6‰ (n = 57) in spring to 13.4 ± 0.5‰ in winter (n = 42), with the highest within-seasonal variation noted in spring (3.1‰) and the lowest in summer (1.8‰; Table 4). Significant differences were observed in δ15N values between spring and winter (p = 0.008), although no significant differences were noted among summer, autumn, and other seasons (p > 0.05; Table 4).

Table 4 δ15N and δ13C values (‰) in the muscle tissue and estimated trophic position (TPgeneral, TPc) of S. commerson specimens across seasons.

Seasons	n	Mean FL ± SD	Mean δ15N ± SD	Mean δ13C ± SD	Mean TPgeneral ± SD	Mean TPc ± SD	
Spring	57	95.31 ± 21.13	13.8 ± 0.6a	−17.3 ± 1.2a,b	4.2 ± 0.2	4.0 ± 0.2	
Summer	22	89.09 ± 5.50	13.6 ± 0.5a,b	−17.6 ± 1.2a	4.1 ± 0.2	3.9 ± 0.1	
Autumn	31	90.87 ± 19.99	13.5 ± 0.6a,b	−17.0 ± 1.2a,b	4.1 ± 0.2	3.9 ± 0.2	
Winter	42	82.93 ± 25.90	13.4 ± 0.5b	−16.7 ± 1.1b	4.1 ± 0.2	3.9 ± 0.1	
Note:

δ15N and δ13C values followed by different superscript letters indicate statistically significant differences among seasons (Kruskal–Wallis test with post hoc multiple comparisons, p < 0.05).

The mean δ13C values were the highest in winter (−16.7 ± 1.1‰, n = 42) and the lowest in summer (−17.6 ± 1.2‰%, n = 22; Table 4). Within-seasonal variation ranged from 3.9‰ in summer to 4.6‰ in spring. Significant differences were observed in δ13C values between summer and winter (p = 0.043). However, no significant differences were noted in δ13C values between the spring, autumn and other seasons (p > 0.05, Table 4).

SIA results for prey species

Figure 6 presents the δ15N and δ13C values for copepods (baseline), semi-digested fish remains, and 22 common prey species. For fish species (including semi-digested fish remains), the mean δ15N values ranged from 7.4 ± 0.6‰ to 12.3 ± 0.8‰ and the mean δ13C values ranged from −18.5 ± 0.1‰ to −16.4 ± 0.4‰. For cephalopods, the mean δ15N values ranged between 10.3 ± 0.1‰ and 11.9 ± 0.4‰ and the mean δ13C values ranged from −18.2 ± 0.1‰ to −17.2 ± 0.2‰. For shrimp species, the mean δ15N values ranged from 8.9 ± 0.3‰ to 9.0 ± 0.2‰ and the mean δ13C values ranged from −15.4 ± 1.1‰ to −14.4 ± 0.1‰.

The mean nitrogen TEF (Δδ15Nanalyzed, Fig. 8A) between common prey species and S. commerson was 3.7 ± 1.5‰. For fish, Δδ15Nanalyzed values ranged from 1.3‰ (Trachurus japonicus) to 6.2‰ (Decapterus macarellus). For cephalopods, Δδ15Nanalyzed values ranged from 1.7‰ (Sepiida) to 3.3‰ (Uroteuthis edulis and Uroteuthis chinensis). For shrimp species, the Δδ15Nanalyzed value was approximately 4.6‰.

Figure 8 Mean trophic enrichment factors (TEFs) for Δδ15Nanalyzed (A) and Δδ13C (B) between common prey species and S. commerson in the central Taiwan strait.

Error bars indicate standard deviation (±SD), sample sizes for each taxon are indicated above the bars. The red dashed line in panel (A) denotes the general trophic enrichment factor for δ15N in marine fish (Δδ15Ngeneral = 3.4 ‰, Vander Zanden & Rasmussen, 1999; Post, 2002).

A comparison of stomach contents (Table 1, Fig. 4) and Δδ15Nanalyzed values across size classes (Fig. 9A) revealed distinct patterns across prey species. In Class I, Δδ15Nanalyzed values ranged from 3.6‰ to 5.2‰ for Sardinella lemuru, D. macarellus, and Decapterus maruadsi. In Class II, the mean Δδ15Nanalyzed values were 1.9‰ for Trichiurus spp. and 2.8 ± 0.3‰ for Uroteuthis spp. In Class III, the mean Δδ15Nanalyzed values were 5.5 ± 1.2‰, 2.4‰, and 3.3 ± 0.3‰ for Decapterus spp., Trichiurus spp., and Uroteuthis spp., respectively. Across Classes IV–VI, the mean Δδ15Nanalyzed values ranged from 5.3‰ to 5.7‰ for Decapterus spp., 2.1‰ to 2.6‰ for Trichiurus spp., and 3.1‰ to 3.5‰ for Uroteuthis spp. Additionally, the Δδ15Nanalyzed value was 4.3‰ for Mene maculata in Class VI. Overall, across all size classes, the mean Δδ15Nanalyzed value for semi-digested fish remains and all common prey species (Class-related Δδ15Nc) ranged from 2.4‰ to 3.7‰ and 2.8‰ to 4.1‰, respectively (Fig. 9A).

Figure 9 Mean trophic enrichment factors for Δδ15Nc (A, ± 1.5 ‰) and Δδ13Cc (B, ± 1.0‰) between common prey species and S. commerson across size classes.

Error bars represent standard deviation. Sample sizes for each prey species are shown in Fig. 6.

The mean δ13C TEF (Δδ13C, Fig. 8B) between common prey species and S. commerson was 0.2 ± 1.0‰. Within prey groups, Δδ13C values ranged from −0.7‰ (Sardinella lemuru) to 1.4‰ (Amblygaster leiogaster) for fish species, from 0.1‰ (U. edulis and Sepiida) to 1.1‰ (U. chinensis) for cephalopods, and from −2.7‰ to −1.8‰ for shrimp species.

In the analysis of the stomach content, Δδ13C values (Fig. 9B) in Class I ranged from −0.2‰ to 1.7‰ for Sardinella lemuru, D. macarellus, D. maruadsi, and semi-digested fish remains. In Class II, the mean Δδ13C values were 1.8 ± 0.5 and 2.3‰ for Uroteuthis spp. and semi-digested fish remains, respectively. In Class III, the mean Δδ13C values were −0.4 ± 0.3‰ for Decapterus spp., 0.4‰ for Trichiurus spp., 0.6 ± 0.5‰ for Uroteuthis spp., and 1.1‰ for semi-digested fish remains. Across Classes IV–VI, the mean Δδ13C values ranged from −0.8 ± 0.3‰ to −0.3 ± 0.3‰ for Decapterus spp., 0.01‰ to 0.5‰ for Trichiurus spp., 0.2 ± 0.5‰ to 0.7 ± 0.5‰ for Uroteuthis spp., and 0.7‰ to 1.1‰ for semi-digested fish remains. Overall, across all size classes, the mean Δδ13C values of all common prey species (Class-related Δδ13Cc) ranged from −0.3‰ to 1.3‰ (Fig. 9B).

TP

Comparison of TP across size classes

Based on Δδ15Ngeneral of 3.4‰, the mean TPgeneral of S. commerson across all size classes was 4.1 ± 0.2 (n = 152, Table 3). TPgeneral significantly increased from 3.8 ± 0.1 in Class I to 4.1 ± 0.2 in Class II, peaking at 4.3 ± 0.3 in Class VI. Within-class TPgeneral variation was greater in Classes IV to VI (0.7–0.8) than in Classes I to III (0.4–0.6; Table 3).

Based on Δδ15Nanalyzed values derived from stomach contents and for common prey species, the Δδ15Nc values across all size classes (~3.72‰) slightly exceeded the Δδ15Ngeneral (3.4‰, Table 3). Consequently, TPc recalculated using Δδ15Nc value for common prey species were analyzed. Mean TPc in Classes I and II remained below 4.0 but exhibited marked increases across size classes (0.1–0.3); the highest mean TPc was observed in Class VI (4.1 ± 0.3). A similar trend was noted for within-class TPc variation, which was greater in Classes IV to VI (0.7–0.8) than in Classes I to III (0.4–0.5). Across all size classes, the mean TPc was 4.0 ± 0.2, which was 0.1 lower than TP (Table 3).

Comparison of TP across seasons

Based on Δδ15Ngeneral of 3.4‰, seasonal variations in the mean TPgeneral of S. commerson aligned with the FLs of the specimens (Table 4). A lower TPgeneral (4.1) was noted from summer to winter, which corresponded to smaller mean FLs. By contrast, the highest TPgeneral (4.2) was observed in spring, which corresponded to the largest mean FLs (Table 4).

When seasonal TPc was recalculated using Δδ15Nanalyzed values for common prey species (Table 4), the highest TPc (4.0 ± 0.2) was noted in spring. The TPc values in the other seasons were approximately 3.9.

Mixing model analysis

An isotopic mixing model, initially incorporating 22 prey sources, retained 14 after two-step filtering; these were used to estimate the mean dietary contributions (%) across size classes (Fig. 10). In Class I, the primary prey items were D. macarellus (12.6%), Sardinella lemuru (12.2%), Etrumeus micropus (10.8%) and Metapenaeopsis barbata (10.2%). In Class II, Metapenaeopsis barbata had the highest contribution (22.6%), followed by Sardinella lemuru (7.7%) and other Penaeidae (7.3%).

Figure 10 Estimated dietary contributions of S. commerson across size classes based on Bayesian isotopic mixing model.

Error bars indicate standard deviation. Sample sizes for each prey species are shown in Fig. 6.

In Class III, major dietary components included Evynnis cardinalis (14.6%), Uroteuthis duvaucelii (10.6%) and Encrasicholina punctifer (8.6%). In Class IV, the diets were mainly composed of Evynnis cardinalis (14.4%), U. duvaucelii (10.9%), U. chinensis (8.2%) and D. macarellus (8.2%). The main prey items were Encrasicholina punctifer (13.2%), U. chinensis (12.1%), U. duvaucelii (12%) in Class V. In Class VI, U. duvaucelii (12.4%), Evynnis cardinalis (12%), Trichiurus spp. (10.5%), Encrasicholina punctifer (10%) were the predominant prey.

Overall, the major dietary contributors of S. commerson showed clear ontogenetic shifts across size classes. In the smaller size classes (I and II), the diet primarily consisted of species from Clupeidae (Sardinella lemuru and Etrumeus micropus), Carangidae (D. macarellus) and Penaeidae (Metapenaeopsis barbata). In intermediate and larger classes (III to VI), key prey included Encrasicholina punctifer, Carangidae (D. macarellus), Evynnis cardinalis, Trichiurus spp., as well as Uroteuthis spp.

Discussion

In this study, we integrated SCA and SIA with an isotopic mixing model to provide comprehensive insights into the foraging ecology and trophic interactions of S. commerson in the waters of the Central Taiwan Strait. These methods were selected considering variations in diet composition, isotope values, and tissue fractionation across species, life stages, and temporal–spatial interactions, as well as the highly migratory behavior and rapid digestion rate of S. commerson (McPherson, 1989).

Given its migratory nature, S. commerson undertakes seasonal north–south migrations between the East China Sea and the Taiwan Strait, primarily driven by temperature shifts and current regimes (Fig. 1A; Lo, 2018; Weng et al., 2020). Such seasonal movements likely affect foraging opportunities and feeding incidence patterns, which are reflected in the present study. While SCA provides a snapshot of recent dietary intake over a short timescale (hours to days), feeding incidence can also be influenced by seawater temperature, seasonal variation, prey availability, and life history stage (Lovell et al., 2024; Poiesz, Witte & van der Veer, 2024).

In the northwestern Pacific, S. commerson spawns between spring and summer. The Taiwan Strait, a major year-round fishing ground for S. commerson (Chen et al., 2021), is also a key spawning area, as indicated by females with hydrated and postovulatory oocytes (Weng et al., 2020). Consequently, smaller individuals (Classes I and II) were predominantly collected during summer and autumn.

As an ectothermic species, S. commerson exhibits higher metabolic rates in warmer waters, which increases feeding demand (Zhang et al., 2018; Wade et al., 2019). Many marine and freshwater fishes also show elevated feeding frequency during prolonged spawning to meet the energetic costs of reproduction (McBride et al., 2015; Torsabo et al., 2022). Consistent with this pattern, feeding incidence across all size classes was higher in warmer months (Figs. S1–S6), likely reflecting both elevated metabolic requirements and increased prey availability during the reproductive season (Shih, Cai & Ni, 2009).

Seasonal variation in prey composition (Fig. 5) was evident: Decapterus spp. were consistently present year-round; Encrasicholina spp. occurred more frequently in spring and winter; Mene maculata was primarily found in spring and summer; Sardinella lemuru was most abundant in summer; and Trichiurus spp. predominated in autumn and winter. The occurrence patterns of Encrasicholina spp. and Mene maculata were closely associated with fishing grounds in the southern Taiwan Strait (Ray et al., 2024; Fisheries Agency, 2025), which overlapped substantially with the winter and spawning migratory routes of S. commerson (Fig. 1A; Lo, 2018; Weng et al., 2020). The high occurrence of Sardinella lemuru in summer coincided with peak catches of other Clupeidae (e.g., Etrumeus micropus) and Engraulidae in Taiwanese waters, mainly harvested by driftnets, purse seines, light fisheries, and trawling nets (Fisheries Agency, 2025); while the abundance of Trichiurus spp. in autumn and winter corresponded with substantial catches in southwestern Taiwan coastal waters using trawling nets, gillnets, set nets, and pole-and-line fisheries (Lai et al., 2020).

Capture-related biases also influenced the results. Feeding patterns may also be shaped by physiological responses to capture (Lovell et al., 2024; Poiesz, Witte & van der Veer, 2024). In this study, specimens were collected using various fishing methods. Class I individuals, which showed feeding incidences exceeding 50% in summer and autumn, were mainly caught by light fisheries and trawling nets. Light fishing, which attracts phototactic organisms, produced the lowest proportion of empty stomachs (Okpala et al., 2017; Xu et al., 2022). By contrast, larger individuals were primarily caught with driftnets, longlines, and gillnets. In these cases, stomach contents were often regurgitated during capture (Sutton et al., 2004), and when combined with the rapid digestion of prey (Chang et al., 2022), these factors likely contributed to the overall high incidence of empty stomachs (79.80%), with the remainder primarily containing semi-digested fish remains. Similar patterns have been reported for S. commerson in Karnataka, India (Rajesh et al., 2017); the northern Great Barrier Reef, Australia (McPherson, 1987); and the Gulf of Thailand (Tongyai, 1970).

Fishing activities and gear type likely influenced prey occurrence in the SCA results. In Taiwan, gillnets are widely used and capture species across a broad range of trophic levels, including Clupeidae, Decapterus spp., and Trichiurus spp., which may be caught alongside Scomberomorus spp. during fishing operations (Wu et al., 2024). Trichiurus spp., Decapterus spp., Mene maculata and small Scombridae are common baits in longline fisheries (Lee et al., 2024), potentially increasing their occurrence in the stomach contents of S. commerson. Decapterus spp., in particular, are a major target in light fisheries, with an estimated annual harvest of 900 mt in the study area (Fisheries Agency, 2025). Other prey items used by commercial fishers, such as Trichiuridae, small Scombridae, Tetraodontidae, and Exocoetidae, may originate from bycatch discards, as reported in northern Australian trawl fisheries (McPherson, 1987). Notably, over 83% of specimens analyzed for SCA in this study were caught by driftnets and longlines, underscoring the potential role of fishing practices in shaping observed diet composition. These observations indicate that the appearance of certain prey species in SCA reflects not only predator size and ontogenetic digestion rates, but also seasonal prey availability and fishing gear type. Moreover, they highlight the importance of pelagic fishes and cephalopods inhabiting shallow waters (<100 m) (Niamaimandi et al., 2015) in the diet of S. commerson, whose availability may decline with increasing depth (Rajesh et al., 2017).

Comparable patterns have been reported in other regions. Finfish and shrimp were identified as key dietary components in the Solomon Islands (Blaber et al., 1990) and along the Egyptian Mediterranean coast (Bakhoum, 2007). A broad diversity of prey items, including crustaceans and cephalopods, has been documented for S. commerson in the waters off Dar-es-Salaam, Tanzania (Johnson & Tamatamah, 2013), and along the eastern coast of Australia, with Exocoetidae, prawns, and squid reported as prominent dietary components (MacInnes, 1950). According to McPherson (1987), S. commerson individuals with FL <50 cm primarily feed on small pelagic fishes, such as species from Carangidae, Clupeidae, Hemiramphidae, and semi-pelagic Leiognathidae, whereas larger individuals (>50 cm FL) consume species from Clupeidae (Amblygaster spp.), Carangidae (Decapterus spp. and Megalaspis spp.), Caesionidae (Caesio spp.), squid and penaeid shrimp. These findings are consistent with the dietary patterns observed in the present study.

Given the rapid digestion rate of S. commerson and the transient nature of its stomach contents, SIA integrates carbon and nitrogen isotopic compositions from prey into predator tissues, providing a more comprehensive understanding of feeding habits. In such analyses, tissue-specific isotopic turnover rates are critical for interpreting foraging patterns over different temporal scales. Muscle tissue, with its relatively low metabolic activity and slower turnover rate, records dietary information over several months rather than weeks (MacNeil, Drouillard & Fisk, 2006; Cerling et al., 2007; Espinoza et al., 2019).

With respect to size-related δ15N values (Fig. 7A), the increase was most pronounced between the smallest classes, whereas subsequent increments in larger classes were relatively minor. These patterns suggest ontogenetic shifts in prey use, as older individuals, less constrained by swimming ability and competitive pressure, can capture higher TPs prey (Bethea et al., 2007; Espinoza et al., 2019). Comparisons between SCA and isotopic mixing model results further indicate that the rise in δ¹⁵N from smaller to intermediate classes is primarily driven by a dietary transition from low TP species (e.g., Sardinella spp., TP ~2.9) to higher TP preys, such as Trichiurus spp., Mene maculata, Decapterus spp., Evynnis cardinalis, and Katsuwonus spp. (TP 3.4–4.0) (Wu et al., 2024). Consequently, larger individuals exploit a broader prey spectrum, resulting in greater trophic niche breadth (Polito et al., 2011; Sánchez-Hernández & Servia, 2012; Weng et al., 2015), as reflected by wider within-class δ15N variation in Classes III to VI.

For δ13C (Fig. 7B), although the values remain relatively stable between prey and predator, they vary among primary producers and habitats, and thus serve as valuable indicators of the basal carbon sources supporting a consumer’s diet. Smaller S. commerson (<70 cm FL; ≤1 year old; Weng et al., 2021), which typically forage and remain around nursery grounds in the central–southern Taiwan Strait (Lo, 2018; Weng et al., 2020), displayed lower within-class variation (1.0–2.0‰). In contrast, larger individuals, with greater swimming capacity and broader migratory ranges spanning the northern South China Sea, the Taiwan Strait, the eastern coast of China, the western coast of Taiwan, and the East China Sea (Fig. 1A; Lo, 2018), exhibited a broader carbon isotopic niche (~4.5‰), reflecting access to more diverse foraging habitats.

Consistent with this pattern, both δ15N and δ13C values exhibited weak but significant correlations with body size (Fig. 7). However, previous studies have reported mixed findings regarding isotopic variation in relation to body size. For instance, Vinagre et al. (2008) found no correlation between δ13C or δ15N values and body size in various fish and cephalopod species. By contrast, Sweeting et al. (2007a, 2007b) reported significant correlations between δ13C values and body size, while Revill, Young & Lansdell (2009) and Varela, Rodríguez-Marín & Medina (2013) observed that δ15N values increased with body size in marine forage fish and predatory species. These discrepancies highlight that isotopic variation results from complex interactions, influenced not only by ontogenetic growth, metabolism, diet composition, and physiological processes, but also by environmental factors such as baseline isotope variability and habitat differences.

In marine ecosystems, phytoplankton serve as the dominant primary producers, accounting for over 99% of total oceanic production and playing a critical role in shaping the spatial and temporal isotopic composition of higher trophic levels, including predators (Horii et al., 2025). Field observations have demonstrated that both δ13C values of dissolved inorganic carbon and organisms tend to decrease with increasing latitude and declining seawater temperature on a global scale (Rau et al., 1992; Goericke & Fry, 1994; Horii et al., 2025). Generally, δ13C values are higher in coastal and inshore regions than in offshore waters (Mitani et al., 2006).

By contrast, δ15N values demonstrate more complex spatial variability, even within similar latitudinal zones, owing to the influence of multiple biogeochemical processes, such as nitrogen fixation, water-column denitrification, and nitrate assimilation by primary producers (Wada et al., 1987; Altabet & Francois, 1994; Horii et al., 2025). In lower-latitude regions, δ15N variation is primarily driven by the relative contributions of different nitrogen sources, including nitrogen derived from atmospheric N₂ fixation and nitrate transported through deep-water upwelling or horizontal advection. In higher latitudes, δ15N values tend to more directly reflect the extent of nitrate utilization (Horii et al., 2025). Generally, oligotrophic waters dominated by N2 fixation exhibit lower δ15N values in particulate organic matter (POM) and phytoplankton, whereas higher δ15N values are typically observed in upwelling regions due to enhanced denitrification. Similar to δ13C, lower δ15N values are commonly observed in offshore waters (Espinoza, 2014; Horii et al., 2025).

These spatial and temporal baseline variations in δ13C and δ15N are ecologically significant, as S. commerson exhibits specialized pelagic feeding behavior (Espinoza et al., 2019) and integrates these signals through thermally driven migrations across multiple coastal and inshore ecosystems of the northwestern Pacific (Fig. 1A). Consequently, seasonal and spatial shifts in the isotopic signatures of primary producers are reflected in the tissue isotopic composition of S. commerson. From spring to autumn, as individuals migrate across the Taiwan Strait and the East China Sea, topographically induced upwelling, monsoonal currents, freshwater discharge during the rainy season (May–June), and typhoon-driven vertical mixing enhance nitrate availability along the coasts of Taiwan and China, the Yun-Chang Rise (central Taiwan Strait), and the Taiwan Bank. These processes likely elevate δ15N values in POM and phytoplankton, which are subsequently reflected in the higher δ15N values observed in S. commerson tissues (Table 4) during these seasons. By contrast, reduced nitrate utilization and lower seawater temperatures in winter contribute to lower δ15N values, despite the southward intrusion of nitrate-rich waters through the China Coastal Current (Huang et al., 2020). In terms of δ13C (Table 4), the highest values were recorded in summer and the lowest in winter, suggesting that seasonal changes in seawater temperature (Figs. S1–S6), rather than latitudinal shifts alone, may be the dominant factor influencing δ¹³C variability in S. commerson throughout its annual migratory cycle.

Compared with SCA, SIA integrated with isotopic mixing models provides comprehensive insights into diet composition, mitigating biases from the high occurrence of empty stomachs and prey-specific digestion rates (MacDonald, Waiwood & Green, 1982). In the mixing model results (Fig. 10), Sardinella lemuru and D. macarellus commonly found in the stomach contents of smaller individuals (Classes I and II), were also identified as major contributors. However, Etrumeus micropus and Penaeidae (Metapenaeopsis barbata) showed relatively higher contributions than indicated by SCA, whereas Trichiurus spp., typically observed in Class II stomachs, accounted for only 5.9% in the mixing model output.

Notable discrepancies between the two methods emerged primarily in larger size classes. According to the mixing model, diet composition was generally similar across Classes III to VI, with Encrasicholina punctifer, Evynnis cardinalis, and Uroteuthis spp. representing the dominant prey items. In contrast, Decapterus spp. and Trichiurus spp., which were abundant in the SCA of larger individuals, contributed notably only in Class IV in the mixing model. Likewise, Mene maculata, frequently observed in the SCA of Class VI, contributed approximately 5–6.5% in the mixing model estimates across intermediate to larger classes (Fig. 4, 10).

These differences may arise not only from opportunistic feeding during fishing operations with bait fish (Sui et al., 2021) but also from the semi-digested state of the stomach contents, which constrains accurate prey identification (Baker, Buckland & Sheaves, 2014). For instance, members of Clupeidae (e.g., Etrumeus micropus, Spratelloides gracilis, Sardinella lemuru, A. leiogaster) and Engraulidae (e.g., Encrasicholina punctifer, Engraulis japonicus) are particularly prone to misidentification when examined in a partially digested condition. In contrast, SIA combined with mixing models integrates assimilated dietary inputs over longer timescales, thereby capturing persistent foraging patterns that extend beyond the short-term snapshots provided by SCA. Moreover, variations in δ13C and δ15N values were evident across seasons and size classes in S. commerson, as well as among prey species, highlighting the complexity of isotopic interpretations. Additionally, small sample sizes for certain prey taxa may have contributed to uncertainty in diet proportion estimates. Therefore, when isotopic mixing models are used to analyze diet composition, it is essential to account for spatiotemporal variation in isotope values, prey-specific trophic discrimination factors, and the size or age structure of both prey and predators to improve the accuracy of diet reconstruction (Noakes & Godin, 1988; Overman & Parrish, 2001; Jennings, Warr & Mackinson, 2002; Munroe et al., 2014; Newsome et al., 2017). Future studies should incorporate larger datasets across multiple seasons and habitats to refine trophic interaction assessments, the results of which can effectively inform management strategies for fisheries.

Variability in δ15N TEFs provides critical insights into TP and species interactions across ecosystems (Hobson & Clark, 1992; Bearhop et al., 2004). These insights are essential for effective management of fishery resources (Caragitsou & Papaconstantinou, 1993). For S. commerson, geographic variations in the breadth of the trophic niche result from differences in baseline isotope values and prey availability. Studies on S. commerson have reported TPs of 4.1 in the Mediterranean Sea (Bakhoum, 2007) and 4.0 in the Persian Gulf (Vahabnezhad et al., 2015). In the present study (Table 3; Fig. 8, 9), the locally derived Δδ15Nc value from prey species was 3.7‰, slightly higher than the commonly assumed value of 3.4‰ (Δδ15Ngeneral). A variation of 1.3‰ in Δδ15Nc was observed across all size classes, corresponding to TPc estimates ranging from 3.8 to 4.3.

In general, δ15N values and TP are expected to increase with body size due to stepwise isotopic fractionation at successive trophic transfers (Revill, Young & Lansdell, 2009; Johnson & Schindler, 2009). However, only a weak correlation between body size and δ15N was observed in this study (Fig. 7A). This pattern likely reflects the opportunistic feeding strategy of S. commerson, which, similar to Japanese Spanish mackerel (Scomberomorus niphonius) (Sui et al., 2021), is a wide-ranging migratory predator that encounters heterogeneous environments and flexibly switches prey according to local availability. Although larger individuals are capable of capturing higher trophic level species, they continue to rely substantially on lower trophic level prey such as Clupeidae. Such dietary generalism dampens the isotopic enrichment that would otherwise be expected from ontogenetic dietary shifts, resulting in relatively modest increases in TP across intermediate and larger size classes.

Wu et al. (2024) investigated historical changes in offshore and coastal fisheries in Taiwan between 1970 and 2021 using mean trophic level and Fishing-in-Balance indices. Their results showed that Scomberomorus spp. occupy a relatively high trophic level (~4.35), consistent with our findings (TP = 3.8–4.3, Table 3), thereby confirming that S. commerson functions as a top predator in the Central Taiwan Strait. As such, fluctuations in its biomass may substantially disrupt ecological stability. Based on dietary composition estimated from both SCA and the isotopic mixing model (Fig. 4, 10), smaller individuals primarily feed on species from Clupeidae, Carangidae (D. macarellus), Penaeidae (Metapenaeopsis barbata) and small Trichiurus spp., whereas intermediate and larger individuals consume a broader range of prey, including Engraulidae, Clupeidae, Carangidae (D. macarellus), Mene maculata, Evynnis cardinalis, Trichiurus spp., and Uroteuthis spp. Because many of these prey species are themselves targeted by commercial fisheries, fluctuations in their abundance may directly shape the feeding ecology of S. commerson.

In recent years, habitat degradation combined with modifications in light fishing and trawling technologies has driven notable declines in catches of Clupeidae (A. leiogaster, Sardinella lemuru, Etrumeus micropus, Spratelloides gracilis), Mene maculata and Penaeidae (Metapenaeopsis barbata, Penaeus japonicus, Parapenaeopsis cornuta). By contrast, landings of Trichiurus spp. have increased significantly since 2020, following the authorization of exports to China through fish carriers. Meanwhile, catches of Carangidae (Decapterus spp., Trachurus japonicus) and Uroteuthis spp. have remained relatively stable over the past 5 years, likely due to regulatory measures implemented for mackerel and torch-light neritic squid fisheries. (Fig. S7, Fisheries Agency, 2025).

These fluctuations in prey availability may influence the feeding strategies of S. commerson across different life stages, potentially altering its foraging behavior and use of nursery grounds around Taiwan. Continual monitoring of spatiotemporal dietary shifts is therefore essential for refining trophic models and ensuring sustainable management of fisheries. Our findings clarify the trophic ecology of S. commerson and its role as a top predator under dynamic environmental and fishing pressures. More broadly, they emphasize the importance of understanding marine predator diets for assessing trophic niches and predicting food web responses to ecosystem change, particularly in the context of global warming, habitat degradation, and overexploitation of fishery resources (Grainger et al., 2020; Tran et al., 2023).

Conclusions

Given the limitations of SCA, which are attributable to the high rate of species-dependent digestion and the transient nature of stomach contents, integrating SCA and SIA with an isotopic mixing model is a valuable approach for reconstructing the foraging ecology of marine species. In this study, we applied this combined method to investigate the diet composition of S. commerson in the Central Taiwan Strait. Overall, 79.80% of specimens had empty stomachs, while the remaining 20.20% contained semi-digested fish remains. The SCA revealed that identifiable prey species varied across size classes. Clupeidae, Decapterus spp., and Trichiurus spp. were prevalent in size Classes I and II, whereas Decapterus spp. and Mene maculata were prevalent in Classes III to VI. Seasonal variations were also evident, with Decapterus spp. consistently present year-round; while Sardinella lemuru and Trichiurus spp. were more frequently detected in summer and autumn–winter, respectively. The isotopic mixing model further highlighted clear ontogenetic dietary shifts. In the smaller size classes (I and II), diets were dominated by Clupeidae (Sardinella lemuru and Etrumeus micropus), Carangidae (D. macarellus) and Penaeidae (Metapenaeopsis barbata). In intermediate and larger classes (III to VI), major contributors included Encrasicholina punctifer, Carangidae (D. macarellus), Evynnis cardinalis, Trichiurus spp., as well as Uroteuthis spp. Differences between prey species identified by SCA and those inferred through isotope analysis likely arise from opportunistic feeding during fishing operations with bait fish, as well as from the semi-digested state of stomach contents, which limits accurate taxonomic resolution. The TP of S. commerson ranged from 3.8 to 4.3 across all size classes, with a mean value of 4.0, confirming its role as a top predator in the Central Taiwan Strait. Fluctuations in its biomass may have significant ecological implications, potentially destabilizing the local marine ecosystem. Overall, our study underscores the value of integrating complementary approaches to improve the accuracy of trophic models and provides essential insights for the sustainable management of fisheries in the region.

Supplemental Information

Supplemental Information 1 Monthly mean sea surface temperature in the northwestern Pacific from January to December 2017.

Data retrieved from the ERDDAP server hosted by the Asia-Pacific Data-Research Center (APDRC), University of Hawaii. Map Source: http://apdrc.soest.hawaii.edu/erddap/griddap/hawaii_soest_f88c_2508_4a21.graph).

Supplemental Information 2 Monthly mean sea surface temperature in the northwestern Pacific from January to December 2018.

Data retrieved from the ERDDAP server hosted by the Asia-Pacific Data-Research Center (APDRC), University of Hawaii. Map Source: http://apdrc.soest.hawaii.edu/erddap/griddap/hawaii_soest_f88c_2508_4a21.graph).

Supplemental Information 3 Monthly mean sea surface temperature in the northwestern Pacific from January to December 2019.

Data retrieved from the ERDDAP server hosted by the Asia-Pacific Data-Research Center (APDRC), University of Hawaii. Map Source: http://apdrc.soest.hawaii.edu/erddap/griddap/hawaii_soest_f88c_2508_4a21.graph).

Supplemental Information 4 Monthly mean sea surface temperature in the northwestern Pacific from January to December 2020.

Data retrieved from the ERDDAP server hosted by the Asia-Pacific Data-Research Center (APDRC), University of Hawaii. Map Source: http://apdrc.soest.hawaii.edu/erddap/griddap/hawaii_soest_f88c_2508_4a21.graph).

Supplemental Information 5 Monthly mean sea surface temperature in the northwestern Pacific from January to December 2021.

Data retrieved from the ERDDAP server hosted by the Asia-Pacific Data-Research Center (APDRC), University of Hawaii. Map Source: http://apdrc.soest.hawaii.edu/erddap/griddap/hawaii_soest_f88c_2508_4a21.graph).

Supplemental Information 6 Monthly mean sea surface temperature in the northwestern Pacific from January to December 2022.

Data retrieved from the ERDDAP server hosted by the Asia-Pacific Data-Research Center (APDRC), University of Hawaii. Map Source: http://apdrc.soest.hawaii.edu/erddap/griddap/hawaii_soest_f88c_2508_4a21.graph).

Supplemental Information 7 Annual catch trends of S. commerson (A), Clupeidae (B), Engraulidae (C), Carangidae (D), Mene maculata (E), Trichiurus spp. (F), Uroteuthis spp. (G), and Penaeidae (H) in Taiwanese waters from 2006 to 2023.

Data source: Taiwan (Fisheries Agency, 2025).

Supplemental Information 8 Raw data for stomach contents analysis of Scomberomorus commerson. .

Supplemental Information 9 Raw data for stable isotope analysis.

Supplemental Information 10 Isotope mixing model code.

Supplemental Information 11 Code for δ 15 N and δ 13 C values comparisons of Scomberomorus commerson specimens across seasons.

Supplemental Information 12 Code for δ 15 N and δ 13 C values comparisons of Scomberomorus commerson specimens across size classes.

We would like to thank Dr. J. J. Shiao, Institute of Oceanography, National Taiwan University for their assistance on stable isotope analysis in this study.

Additional Information and Declarations

Competing Interests

The authors declare that they have no competing interests.

Author Contributions

Li Chi Cheng conceived and designed the experiments, performed the experiments, analyzed the data, prepared figures and/or tables, authored or reviewed drafts of the article, and approved the final draft.

Jia Shin He conceived and designed the experiments, performed the experiments, analyzed the data, prepared figures and/or tables, and approved the final draft.

Chi Chang Lai performed the experiments, analyzed the data, prepared figures and/or tables, and approved the final draft.

Yen Hung Lee performed the experiments, analyzed the data, prepared figures and/or tables, and approved the final draft.

Jinn Shing Weng conceived and designed the experiments, performed the experiments, analyzed the data, prepared figures and/or tables, authored or reviewed drafts of the article, and approved the final draft.

Hsing Han Huang performed the experiments, analyzed the data, prepared figures and/or tables, and approved the final draft.

Yi Shu Wu performed the experiments, analyzed the data, prepared figures and/or tables, and approved the final draft.

Animal Ethics

The following information was supplied relating to ethical approvals (i.e., approving body and any reference numbers):

Ethical approval was not required because the study did not involve experimentation on live animals.

Field Study Permissions

The following information was supplied relating to field study approvals (i.e., approving body and any reference numbers):

Samples were collected monthly from the Penghu fish market. Fish were caught using drift nets, trolling lines, light fisheries, trawling nets, or longlines by vessels operating in the Central Taiwan Strait. After capture, specimens were immediately stored on ice and transported to the laboratory for analysis.

Data Availability

The following information was supplied regarding data availability:

The raw data is available in the Supplemental Files.

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
