# Peer review of "Feeding ecology and trophic interactions of the narrow-barred Spanish Mackerel (Scomberomorus commerson) in the Central Taiwan Strait"

_PeerJ, doi:10.7717/peerj.20350_

## Round 0.1 · original submission · Major Revisions

· Academic Editor

Major Revisions

Dear Dr. Cheng,

You can find the comments and suggestions of the expert reviewers in the attached reports. As you will see, expert reviewers have pointed out the critical errors. Consequently, a major revision is needed for your article.

I request you that improve your manuscript following the reviewers' suggestions

Sincerely

Reviewer 1 ·

Basic reporting

There are no problems of English and article structure in this paper, as far as I could check. Trends of Scomberomorus commerson in other areas are also cited, indicating a broad view of this species. As a supplement information, Shoji et al. (1997) describes diet compositions of S. niphonius during larval and juvenile stages.

However, there are some points to improve on this paper.
First, the paper presents most of the results in tables, but it seems to be better that the results are presented in figures as well as tables for understandings. (MixSIAR results, what is shown in the Table, etc.)
On the other hand, there is no Table of sample composition. Table of samples (both target species and prey species) should be prepared to make it easier to understand that so many samples were analyzed.
In addition, a table of seasonal and size composition by fishing method would be helpful to understand this paper.

Reference in this comment
Shoji, J., Kishida, T., and Tanaka, M (1997) Piscivorous habits of Spanish mackerel larvae in the Seto Inland Sea. Fish. Sci., 63, 388-392.

Experimental design

The paper seems to be robust in that it not only examined the stomach contents of a very large number of samples over a period of 5 years, but also looked at long-term trends using muscle stable isotope ratio analysis, both in S. commerson and in candidate prey species. The importance of investigating diet composition of fisheries resources seems to be well defined.

However, I would like to know why extractions of lipid from dorsal muscles were not conducted in this paper. It was reported that lipid in muscles potentially caused the difference of the carbon stable isotope ratio values (DeNiro and Epstein, 1977). Therefore, I recommend the addition of some sentences which explained the reasons, or the correction of the value of carbon stable isotope ratio such as Logan et al. (2008), in case of C:N ratios obtained.
I could not find a description of which value of discrimination factor is used for MixSIAR analyses. It is better to explain how refer or calculate this value in the Materials and Methods section.
In addition, I didn't understand how to obtain TPc. I would like to see more specific information in the Materials and Methods section.

References in this comment
DeNiro, M. J., and S. Epstein. 1977. “Mechanism of Carbon Isotope Fractionation Associated With Lipid Synthesis.” Science 197: 261–263. https://doi.org/10.1126/science.327543
Logan, J. M., T. D. Jardine, T. J. Miller, S. E. Bunn, R. A. Cunjak, and M. E. Lutcavage. 2008. “Lipid Corrections in Carbon and Nitrogen Stable Isotope Analyses: Comparison of Chemical Extraction and Modelling Methods.” Journal of Animal Ecology 77: 838–846. https://doi.org/10.1111/j.1365-2656.2008.01394.x.

Validity of the findings

The objectives and conclusions are clear and well stated throughout the paper.
However, I could not find the comparison with environmental factors such as denitrification and nitrogen fixation overall. If there are previous studies in the relevant area, please indicate them. In addition, Ohshimo et al. (2019) showed that the dependency of carbon stable isotope ratio on temperature.
There is a discussion of sustainable resource management in the Introduction and Conclusion section, but there is no specific connecting results in Results and Discussion section. If there are sentences that can be described in connection with resource management, for example, citing the stock status of one of the major prey species such as Decapterus spp. or Uroteuthis spp., it is recommended to add the discussion.

Reference in this comment
Ohshimo, S., D. J. Madigan, T. Kodama, et al. 2019. “Isoscapes Reveal Patterns of δ13C and δ15N of Pelagic Forage Fish and Squid in the Northwest Pacific Ocean.” Progress in Oceanography 175: 124–138.

Additional comments

L170, 201: The order of the explanations does not match the order of Materials and Methods. Please list W, N, F, and IRI in this order so that readers can see the results together with the Table.
L358: What is the definition of “Juveniles”? I recommend citing any paper on growth maturity of S. commerson.

Reviewer 2 ·

Basic reporting

(1) The manuscript is written in clear, unambiguous, and professional English throughout, adhering to the standards of an international journal. The introduction provides sufficient background and a thorough literature review, offering appropriate scientific context. The structure of the manuscript is professional, and all figures, tables, and raw data are presented transparently and support the research findings.
(2) I suggest reducing repetitive phrases in the abstract and splitting long sentences for improved readability. Some paragraphs in the introduction could be divided for better clarity, and the knowledge gap could be articulated more explicitly.
(3) Figures and their legends are generally well prepared. However, adding the sample size (n) directly to the figure legends would enhance data interpretation.

Experimental design

the journal. The research question is clearly defined, and the authors explicitly state how their work addresses a specific knowledge gap regarding the feeding ecology of Scomberomorus commerson in the Taiwan Strait.
(2) The experimental design is rigorous, with a large sample size collected over multiple years, stratified by size and season, and employing both stomach content analysis and stable isotope analysis. The methods are described in sufficient detail to allow replication.
(3) I recommend providing more explicit information regarding some statistical procedures (e.g., details of the Kruskal-Wallis test and specific parameters/settings in the MixSIAR model) to further enhance reproducibility. It would also be beneficial to acknowledge study limitations, such as possible seasonal sampling biases and the high proportion of empty stomachs, which may affect data interpretation.

Validity of the findings

(1) The data are robust, with adequate sample size and transparent sharing of underlying datasets. The analyses are statistically sound and well controlled, employing appropriate models and tests.
(2) The conclusions are well stated and are closely linked to the original research questions, with interpretations limited to what is supported by the results.
(3) I suggest that the authors further discuss potential sources of bias, such as the high percentage of empty stomachs and possible limitations due to the sampling area or period, and consider how these might impact the findings. Suggesting ways to address these issues in future studies would enhance the manuscript’s value.

Additional comments

The manuscript addresses a relevant regional and ecological topic, and the overall design and analysis are rigorous and suitable for publication in PeerJ. Minor revisions according to the suggestions above would further improve the quality of the article.

---

## Round 0.2 · Major Revisions

· Academic Editor

Major Revisions

Dear Dr. Cheng,

According to the expert reviewers' evaluation, there are some parts of your valuable article that need to be improved. You can find the comments and suggestions of the expert reviewers in the attached reports. As you will see, expert reviewers have pointed out some errors. Consequently, a major revision is needed for your article.

I request that you improve your manuscript following the reviewers' suggestions.
Sincerely,

Reviewer 1 ·

Basic reporting

Significant revisions made the paper very clear and substantial. There are some points to modify to improve this paper as below:

Figure1: It would be better to illustrate migration information of Scomberomorus commerson to Figure 1.
Figure2: It was hard to understand where feeding incidences were shown. It is recommended to add this information to sample sizes above each bar.
Figure8: It would be better to add explanation about Δδ15Ngeneral in the caption.
Table3,4: It should show which values have significant differences between size class and season not only in the sentences but also in Tables.

Experimental design

This paper was greatly enriched by the addition of information on environmental conditions, statistical methods and prey species. There are enough investigations to discuss the research question defined in this paper.

Validity of the findings

The necessary information is illustrated, and the argument is very clear. However, there are some points to be considered.

When considering seasons, it is recommended below:
・Which season is considered as the starting point of their life history? It is recommended to include the reproductive relationships of the species.
・Consider the effect of turnover (how long is the turnover of the species?)

The specimens of Class I were caught by light fishery and trawling net in summer and autumn. Therefore, it should consider the reason why Sardinella lemuru appeared as stomach contents. In other words, the appearance of this species potentially depends on not only sizes of predators but also seasonal effects or gear type.

This paper has revealed feeding habits of S. commerson, considering the landings of prey species. It would be better to add information on feeding patterns of S. commerson as predators (i.e. opportunistic or selective) to assess the sustainable management.

L387: Concrete temporal information about rapid digestion rate would be necessary if there is information in the reference.
L497: Has Etrumeus micropus referred in SCA? Apparently, this species appeared only in SIA.
L61-62, L536: It is confusing to understand whether S. commerson is top predator or not. Please clarify this information.

Additional comments

L212, 220, 230,L244: Please cite figures or references in the first sentence of each paragraph, such as L299.
L394-395: It seems that this information on phototactic organism, should be written with reference.
L468-479: It would be better to move this paragraph to Introduction section and visualize migration pattern explained in this paragraph in Figure 1.
L542-551: Please cite figures or references in the first sentence of each paragraph.

Reviewer 2 ·

Basic reporting

The manuscript is clearly written, and the English is appropriate for conveying the scientific content. The introduction provides sufficient context and references relevant literature to support the background. The article is well structured, and the figures and tables are professionally presented. Sample sizes are now included in the figure legends, improving transparency. The methods and raw data are clearly described. The results are relevant and directly address the research questions, and the manuscript presents a complete and coherent analysis.

Experimental design

The study is based on a comprehensive field dataset collected over several years and across multiple seasons, with an adequate sample size. The methods, including stomach content analysis, stable isotope analysis, and isotopic mixing models, are appropriate and well explained. Statistical procedures and model parameters are clearly described, which supports reproducibility. The experimental design is suitable and aligns with the research questions.

Validity of the findings

The findings are well supported by the data and statistical analysis. The authors recognize and discuss potential limitations, including the number of empty stomachs and seasonal sampling constraints. The conclusions are clearly based on the results and are presented without overinterpretation. The study provides a reliable and meaningful contribution to the field.

Additional comments

The authors have responded thoroughly and thoughtfully to all reviewer comments. The manuscript has been substantially improved through careful revision and is now ready for publication. I have no further suggestions for improvement.

---

## Round 0.3 · Minor Revisions

· Academic Editor

Minor Revisions

Dear Dr. Cheng,

Based on the expert reviewers' assessment, some aspects of your valuable article need improvement. You will find the reviewers' comments and suggestions in the attached reports. As you will notice, the reviewers have identified some errors. Therefore, a minor revision is required for your article.

I kindly ask that you revise your manuscript according to the reviewers' suggestions.

Sincerely,

Reviewer 1 ·

Basic reporting

no comment

Experimental design

no comment

Validity of the findings

The authors have responded to all requests from reviewers.
By sincerely responding many comments, this paper became a comprehensive examination of Scomberomorus commerson in the Central Taiwan Strait, covering not only its feeding habits but also its life history, environmental responses, and sustainable management.

Additional comments

I have no further revision requests except for the points below.
L442-443: This part states that Encrasicholina spp. appears most frequently in spring and autumn, but looking at Figure 5, this species appears to be spring and winter. Please check, and correct if necessary.
L667: Space is needed between "integrating" and "SCA".

---

## Round 0.4 · accepted · Accept

· Academic Editor

Accept

Dear Dr. Cheng,

I thank you for making the corrections and changes requested by the reviewers. I read and checked your valuable article carefully and am happy to inform you that the article has been accepted for publication in PeerJ.

Sincerely yours,

Reviewer 1 ·

Basic reporting

no comment

Experimental design

no comment

Validity of the findings

no comment

Additional comments

The authors have responded to all requests from reviewers.
This paper shows comprehensively feeding habits, life history, environmental responses, and sustainable management of Scomberomorus commerson, by sincerely responding many comments.
I have no additional revision requests. I recommend that this paper be accepted.